# Biomechanical Rupture Risk Assessment in Management of Patients with Abdominal Aortic Aneurysm in COVID-19 Pandemic

**DOI:** 10.3390/diagnostics13010132

**Published:** 2022-12-30

**Authors:** Lubos Kubicek, Radek Vitasek, David Schwarz, Robert Staffa, Petr Strakos, Stanislav Polzer

**Affiliations:** 12nd Department of Surgery, St. Anne’s University Hospital Brno and Faculty of Medicine, Masaryk University, 612 00 Brno, Czech Republic; 2Department of Applied Mechanics, VSB-Technical University of Ostrava, 708 00 Ostrava, Czech Republic; 3IT4Innovations, VSB—Technical University of Ostrava, 708 00 Ostrava, Czech Republic

**Keywords:** abdominal aortic aneurysm, biomechanics, rupture risk, predictability, COVID-19

## Abstract

Background: The acute phase of the COVID-19 pandemic requires a redefinition of healthcare system to increase the number of available intensive care units for COVID-19 patients. This leads to the postponement of elective surgeries including the treatment of abdominal aortic aneurysm (AAA). The probabilistic rupture risk index (PRRI) recently showed its advantage over the diameter criterion in AAA rupture risk assessment. Its major improvement is in increased specificity and yet has the same sensitivity as the maximal diameter criterion. The objective of this study was to test the clinical applicability of the PRRI method in a quasi-prospective patient cohort study. Methods: Nineteen patients (fourteen males, five females) with intact AAA who were postponed due to COVID-19 pandemic were included in this study. The PRRI was calculated at the baseline via finite element method models. If a case was diagnosed as high risk (PRRI > 3%), the patient was offered priority in AAA intervention. Cases were followed until 10 September 2021 and a number of false positive and false negative cases were recorded. Results: Each case was assessed within 3 days. Priority in intervention was offered to two patients with high PRRI. There were four false positive cases and no false negative cases classified by PRRI. In three cases, the follow-up was very short to reach any conclusion. Conclusions: Integrating PRRI into clinical workflow is possible. Longitudinal validation of PRRI did not fail and may significantly decrease the false positive rate in AAA treatment.

## 1. Introduction

The pandemic infection of the new coronavirus struck the whole world in 2020 and had a major effect on healthcare systems worldwide. During the first wave in spring 2020, governments in many countries ordered the postponement of most elective surgeries, in order to increase the number of available beds in intensive care units (ICU) for the needs of patients with COVID-19 [1]. Typically, this also affected the indication process and treatment planning for patients with abdominal aortic aneurysms (AAA) [2].

AAA is a local, gradual growth and permanent dilatation of the abdominal aorta. It is mostly asymptomatic and its main danger lies in the fact that it can rupture, which is a life-threatening event with a mortality of about 50% even in developed countries [3] (should a patient be admitted to a specialized vascular center; otherwise, the mortality is even higher). On the other hand, not all AAAs rupture. It was observed that the rupture rate largely depends on its maximal diameter; the annual risk of rupture for AAAs < 55 mm is 0.1–1% [4], 55 mm < AAA < 70 mm is about 10% [5], and AAAs ≥ 70 mm is about 30% [5,6]. Therefore, guidelines based on these observations [7] propose intervention when the maximal diameter of AAA exceeds 55 mm (50 mm for women), if the AAA is symptomatic (mostly abdominal or lumbar pain) or if it grows faster than 5 mm in 6 months. Nevertheless, this practice is associated with very low specificity, and the majority of patients are not under imminent risk of rupture when operated on [4,5,6]. Moreover, a significant portion of AAAs will never rupture and are found to be an incidental observation during an autopsy of patients who died due to another condition. Therefore, there is ongoing intensive research to increase the specificity of AAA treatment. One of the most promising directions is biomechanical rupture risk assessment (BRRA) [8,9]. BRRA considers an AAA as a pressure vessel and estimates its risk of rupture via some specific metric, such as peak wall stress or some ratio of wall stress [9] to wall strength [8]. These models have been under development for more than two decades. Recently, a blinded retrospective study [10] has shown that BRRA using an advanced metric called probabilistic rupture risk index (PRRI) is superior in specificity (correctly predicting stable AAA) and has the same sensitivity (correctly predicting AAA ruptures) when compared to the maximum diameter criterion up to 9 months ahead. PRRI is a value stated in percentages and informs us about the probability of rupture during the time period, in which the AAA geometry did not significantly change (roughly a year).

An unprecedented situation created by the COVID-19 pandemic caused the postponement of elective open surgery operations as well as the endovascular repair (EVAR) of AAAs, which would have been operated on under normal situations (AAAs with a maximal diameter of ≥55 mm). EVAR is suggested as a preferred method for emergency repair of ruptured AAAs [11,12] and for COVID-19 positive intact patients [13]. Some hospitals developed a “Level of Priority” to enhance cardiovascular surgery guidelines for choosing patients and divided them into four categories (elective/urgent/emergent/salvage cases) [14]. To some extent, this is supported by the numerical risk analysis [2] but it inevitably leads to an increase in ruptured AAAs. Moreover, the fear from COVID-19 among patients caused an increase in patients with AAA who refused operation (see Table 1). There was a significantly higher number of patients waiting for an AAA operation after the end of the state of emergency. In this situation, we decided to further validate our BRRA approach in a longitudinal study and to investigate the feasibility of incorporating BRRA of AAA into the daily medical workflow in the clinical center of vascular surgery.

## 2. Materials and Methods

### 2.1. Patient Acquisition

The state of emergency was effective in the Czech Republic for 66 days between 12 March and 17 May 2020. The study was conducted at St. Anne’s University Hospital in Brno, Czech Republic. This hospital was dedicated by the ministry of health as one of the key hospitals to care for COVID-19 positive patients. Moreover, the government measures included the postponement of elective operations and affected the treatment of patients with AAA. During this period, a previously planned elective open AAA repair (OAR) was canceled for seven patients in our center. Furthermore, additional 12 patients were registered as suitable for elective AAA repair; however, the date of open AAA repair was not established due to the state of the COVID-19 emergency, which led to the postponement of the required surgery by at least 2 months. Consequently, there was a queue of 19 patients with AAA indicated for intervention when the state of emergency ceased. Typically, all these patients cannot be operated on immediately due to hospital capacity, which led to an increase in the need for alternative AAA rupture risk assessment methods regarding the alignment of patients on the waiting list. Moreover, nine of these patients refused the intervention during the COVID-19 restriction period (six of them due to a fear of the COVID-19 situation in Czech hospitals). All patient data used in this study were the standard of hospital care data. Patients did not sign any additional consent form and Ethical Committee approval was not considered necessary since all patients signed a standard scientific consent form at the beginning of their treatment in St. Anne’s University Hospital allowing for the use of their standard data for scientific purposes.

### 2.2. Study Design

The computational part of the study was conducted between 25 May and 18 July 2020 with a follow-up until 10 October 2020. To investigate the feasibility of cooperation between the biomechanical and medical teams, we proposed a modified AAA management workflow as depicted in Figure 1.

As this type of analysis is not carried out very often, the biomechanical team is not part of the hospital crew. Members of the biomechanical team are situated at VSB Technical University of Ostrava and all communication between the medical and biomechanical teams can be easily performed online. The cost-efficiency of this type of cooperation is yet to be assessed.

Anonymized computed tomography-angiography scans (CT-A) of 19 cases who satisfy the criteria for intervention were referred to the biomechanical team, along with patients’ blood pressure and the date of their CT-A examination. If several blood pressure values were available for one patient, then they were all obtained in the analysis; however, for the majority of patients, only one value was available. Moreover, there were cases of patients referred from elsewhere with only CT scans and no blood pressure information. In this situation, the average value from other patients was used during the analysis. Absence of valid blood pressure may be a significant limitation in PRRI estimation if its obtained value by means of cohort pressure is close to the safe/risky threshold. Anonymization was performed by the medical team who replaced patient-specific information by numbers. Typically, they maintained the ability of inverse identification of individual patients that were referred to the biomechanical team. Clinical data of patients included in the study are reported in Table 2.

The cohort contained 19 patients (5 females and 14 males). In our study, none of the patients suffered from COVID-19 infection and the maximal diameter range was $D_{max}\in \langle 49-83\rangle \;\mathrm{mm}$. The PRRI was computed by the biomechanical team as described in Section 2.2. The whole process of one case of computation was expected to last for 3 days, and the medical team operated independently unless it received a warning regarding patients with high PRRI from the biomechanical team. The medical team scheduled individual cases in the study, and the other cases that appeared before all of the queued patients were operated on according to the current guidelines—cases were ranked and operated on according to a sex-adjusted maximal diameter $D_{sex}$. Moreover, the time that had passed from the originally planned date of intervention was taken into consideration (indicating that longer waiting patients with larger AAAs were scheduled first). The $D_{sex}$ is defined as follows:$$D_{sex}=D_{max}\;\left[\mathrm{mm}\right]\;in\;males\;\;\;or\;\;\;\;D_{max}+5\;\left[\mathrm{mm}\right]\;in\;females$$

All AAAs in these 19 cases remained asymptomatic during the COVID-19 period. The biomechanical team contacted the medical team immediately only in cases of identification of patients with a high risk of AAA rupture (PRRI > 3%). Then, if the patient was still in a queue, the priority in AAA intervention was offered by the medical team. Our center performs primarily open aneurysm repair (OAR) (EVAR is performed in about 15% of patients per year).

Patients who were not operated on (since they refused intervention) were followed until October 2021 by an in-office visit in our center (usually with an ultrasound examination), by a phone call with the patient or by confirming their survival with their general practitioner. The actual length of follow-up (from the date of CT-A) can be found in Table 3.

### 2.3. Biomechanical Rupture Risk Assessment via PRRI

The biomechanical team reconstructed patient-specific geometry of individual AAA cases using the in-house plugin BDICOM for open-source software Blender [15]. In addition, the reconstructed geometry was exported into Ansys ICEM CFD 2020R1 (Ansys Inc., Canonsburg, PA, USA), where a finite element mesh of the vascular wall and intraluminal thrombus was created. This approach does not assess the peak wall stress from Laplace law; rather, the finite element (FE) method divides any geometry into a large number of small FEs. All FEs are connected together and create the FE mesh, which is used to compute stresses and strains in the whole AAA. The high-fidelity level of the FE analysis was used to calculate the peak wall stress in the patient-specific aneurysm. Briefly, our approach has the following key features: (i) Patient-specific geometry of both the aneurysmal wall and intraluminal thrombus (ILT); (ii) a realistic non-linear constitutive model of the aneurysmal wall [16,17]; (iii) realistic constitutive model of ILT respecting the gradual decrease in mechanical properties with the increasing distance from the luminal side [18]; (iv) population-based distribution of wall thickness (obtained during previous studies including histological and mechanical examination of AAA wall) [19]; (v) respects zero pressure geometry of AAA (the zero pressure geometry should be used when the realistic material model is used, without it the deformed geometry would not be the one we see on CT, thus it would not be realistic) [8]; (vi) respects residual stresses in aneurysmal wall [20]; (vii) peak wall stress evaluated when AAA is overloaded by 1.5× patient-specific mean arterial pressure [8]; (viii) PRRI is calculated by considering the population-based distribution of wall strength [8,21]. This approach was described in detail and validated elsewhere [8,10]. In general, PRRI is designed to estimate the probability of AAA ruptures when overloaded by high blood pressure (1.5× patient-specific mean arterial pressure) which is, for the modeling purpose, assumed not to be exceeded in a year. It is a necessary simplification since a more precise estimation of maximal blood pressure that the patient will experience in a year requires significantly more advanced statistical methods and, most importantly, patient-specific blood pressure Holter measurements [22], which was not available here. Consequently, in theory, PRRI should represent the annual risk of AAA rupture.

Identification of AAAs with high risk of rupture via PRRI was validated in a previously published blind study, where it was shown to be superior over $D_{sex}$ in predicting AAA ruptures up to 9 months ahead [10]. Specifically, PRRI was able to decrease the ratio of false positive cases to one third compared to $D_{sex},$ but its power decreases with the increasing time from the initial CT-A. This is due to the fact that PRRI evaluates the risk of rupture only at the baseline and AAA, in which growth is not considered. This is why we applied it first to cases with most recent CT-A (not older than 9 months). Nine months is the limit until which PRRI gives a more accurate prediction than $D_{sex}$ [10]. Cases with CT-A older than 9 months served mainly for further validation of our previously estimated thresholds for safe (PRRI < 3%) and risky (PRRI > 3%) AAA cases. Any case marked as stable (PRRI < 3%) and yet experienced an AAA rupture was considered as false negative from the analysis point-of-view. Similarly, cases were considered as false positive if an AAA was marked as high risk (PRRI > 3%) and confirmed as non-ruptured for at least 6 months from CT-A.

## 3. Results

BRRA analysis was performed in 19 cases of AAA during the first COVID-19 emergency period in our center. Of these, twelve patients had CT-A performed less than 9 months before the assessment. Two patients (2) had a second CT-A scan during the follow-up period—growth rate for the first was 10 mm/year, for the second it was 7.6 mm/year. All other patients had only one baseline diameter measurement. Results for all cases can be found in Table 3. The mean time for computing one case was 2.8 days. Twelve cases were identified as safe (PRRI < 3%), while six cases were identified as high risk (PRRI > 3%) and one case was identified as a very high-risk case (PRRI > 10%). PRRI for this case was 21.3%. Three high-risk patients had already received open AAA repair when PRRI was calculated, while priority in open AAA repair was offered to the other four patients. Unfortunately, none of them underwent a proposed intervention. Three patients due to fear from COVID-19 infection or a fear of operation itself, and a fourth patient died in May 2020 due to sepsis (caused by broncho-pneumonia) with no AAA symptomatology. Open surgery was performed on all (10/19) operated cases (52.6%), while nine patients refused operation (47.3%)—five of them due to fear from COVID-19 infection. One of the refusing patients underwent the repair after the COVID-19 emergency ceased, while the other eight patients still refuse the repair and are followed; all of them are alive at the time of the paper’s creation (August 2021). One patient died within 30 days after open AAA repair (OAR).

The mean follow-up time from CT-A was 459 days (96÷980 days). According to our hospital database, none of the patients included in this project have been infected with SARS-CoV-2 during the follow-up. The largest safe case had $D_{sex}=87\;\mathrm{mm}$, while the smallest high-risk case had $D_{sex}=59\;\mathrm{mm}$. Both mentioned cases are depicted in Figure 2. No prediction was considered as false negative, while 33% of cases (6/18) were false positive. In one case, the follow-up time was considered as very short to reach any conclusion. Finally, no correlation between $D_{sex}$ and PRRI was observed (r = 0.1, *p* = 0.69).

## 4. Discussion

In this study, we tested the feasibility of incorporating BRRA in the AAA decision-making process during the era of the COVID-19 pandemic. This unprecedented situation caused an accumulation of patients indicated for AAA repair in our vascular center as a consequence of the declared state of emergency in the Czech Republic, leading to a postponement of all elective surgery by more than 2 months. Under these circumstances, we tested all postponed AAA cases using BRRA to assist in identifying cases with high risk of AAA rupture. It is important to note that PRRI was previously shown to have a significantly weaker correlation with $D_{sex}$ compared to alternative approaches [10]. In this dataset, we did not even observe a significant correlation between $D_{sex}$ and PRRI (Pearson’s correlation coefficient r = 0.1, *p* = 0.69). In addition to a small cohort, the explanation of this difference may be related to the fact that this cohort was not biased by case selection for the first time. In a regular situation, cases with $D_{sex}>55\;\mathrm{mm}$ are mostly operated on, while in this study, they were followed for a reasonable period of time. This was not previously possible for ethical reasons. Furthermore, this shows that PRRI respects different patient-specific features of AAA (general morphology, ILT, and blood pressure) which results in a more accurate prediction of imminent AAA ruptures [10] and creates an opportunity for a patient-specific indication approach.

### 4.1. BRRA via PRRI Can Distinguish Safe and High-Risk AAAs

Results showed that the chosen PRRI threshold for the identification of stable AAAs resulted in high sensitivity with no confirmed false negative cases (marked as safe and yet experienced a rupture) and the mean follow-up time of stable cases was 459 days (96÷980 days). However, the assessment of specificity is significantly less robust. Six cases were considered as false positive (marked as high risk yet remained stable for at least 4 months) with a mean follow-up time of 685 days (492÷959 days). In one case, the follow-up time was very short (89 days) to reach any conclusion. This results in a false positive rate of 33% (4/16), with about 61% less false positive cases produced via PRRI compared to $D_{sex}$ criterion, which suggested intervention in 94% (17/18) cases. Decrease in false positive rate is extremely important in the era of the COVID-19 pandemic when there is a great need to reduce the number of elective operations to free hospital capacity for COVID-19 patients. This occurred in Italy and Spain in spring 2020 and, since then, it was repeatedly a reality in many countries.

Typically, it is possible that AAAs who identified as stable and safe at the time of analysis can grow over time and the PRRI value can increase. Moreover, these patients would require the AAA repair in the future, but this does not hold for many patients as only 25% [23] to 50% [5] of unoperated AAAs rupture. Therefore, many patients will likely remain in the safe group for their lifetime and may never require the AAA repair. However, a larger group with longer follow-up is required to verify BRRA, which can identify them properly.

Furthermore, it is worth mentioning that having this diagnostic instrument is very useful in emergency situations, such as COVID-19 pandemic or even in a situation when the patient has other severe conditions (e.g., cancer) and we have to decide whether his/her AAA is safe enough for other conditions to be treated first or if there is imminent risk of rupture. As shown in our group, a significant portion of patients refused the proposed surgery. In this case, the BRRA analysis can be used as another instrument on how to convince the patient to undergo AAA repair (high risk cases).

Nevertheless, the above performed assessment of sensitivity and specificity is mostly related to the chosen threshold between the safe and high-risk cases and does not indicate that PRRI failed three times. It is aimed at mimicking the annual risk of rupture, thus it is perfectly reasonable to expect no ruptures even in cases where PRRI equals approximately 20%. Statistically speaking, the overwhelming majority of cases (80%) with this PRRI should remain intact for 1 year. It was considered as false positive only since in a regular situation the sensitivity is significantly more important than specificity (and the guidelines reflect that clearly [7]). Thirty-day mortality associated with elective surgery is no more than 5%, [24] while mortality associated with ruptures is about 50% [3] and even more if a patient is not admitted to a vascular center. From this point-of-view, this partial longitudinal validation of PRRI did not fail. Furthermore, it is worth mentioning that despite this, our aim is to compare the PRRI percentage values with AAA rupture risk, which is normally stated in the literature based on maximal diameter (or mortality risk of elective surgery), such as validation that is not finished. Therefore, it is better to interpret these values as thresholds for three groups of patients according to the estimated rupture risk—safe (patient should be followed), high risk (AAA should be repaired electively), and very high risk (AAA should be repaired as soon as possible), e.g., from a clinical point-of-view, there is no significant difference between the PRRI values of 15% and 60% and rupture risk is almost the same. Finally, no refinement of the PRRI threshold could be carried out due to the absence of ruptured AAAs and small cohort.

### 4.2. BRRA via PRRI Offers Less Compromises in the Pandemic Era

At the time this study was designed, only a few studies regarding AAA management in the pandemic era were published [25]; therefore, we can compare the proposed workflow only due to the ex-post evaluation. Guidelines from the American College of surgeons suggest the postponement of elective operations even for large AAAs if possible in the environment of significant probability of COVID-19 infection [26]. This suggestion is based on the fact that older patients have significantly worse outcomes if infected by COVID-19 [27]. The suggested approach would inevitably be associated with a significant decrease in sensitivity leading to considerably more AAA ruptures (estimated 5% in 60-month deferral). On the contrary, BRRA can assist in postponing two thirds of interventions on a patient-specific basis. Moreover, this study as well as previous results [10] suggest that this approach would be necessary for compromising the sensitivity of interventions.

Another study suggests choosing EVAR over open surgery as the primary operational technique during the COVID-19 pandemic [2]. This approach has a clinical and logical foundation as patients after EVAR are less likely to require ICU care with mechanical ventilation than patients after open repair. Unfortunately, this option is not possible in all cases due to morphological AAA characteristics or technical and financial reasons. More importantly, the incorporation of BRRA in the decision-making process is not in conflict with this suggestion since BRRA never interferes with the decision of how to operate, except only when intervention is necessary. Furthermore, this study suggests the postponement of interventions in large AAA cases (up to 70 mm) of older patients.

### 4.3. Proposed Workflow Is Generally Clinically Applicable

In addition to PRRI validation, we tested the practical cooperation between the surgical and biomechanical teams. Prior to this study, there were concerns regarding the practical applicability of BRRA which requires about 3 days to evaluate a single patient [28]. No concern was recorded from the surgical team on this topic since interventions are planned in our center typically about 30 days after AAA CT-A assessment. On the other hand, if the BRRA should be included in the decision-making process, it is suggested to avoid delays between CT-A acquisition and sending the data to the biomechanical team since the PRRI estimates risk of rupture at the time of CT-A and its power decreases over time. Therefore, for example, if there is a 4-month delay between CT-A acquisition and sending the data to the biomechanical team, the computed PRRI is related to the past 4 months (where it has no clinical relevance) and a maximum of 5 upcoming months. Furthermore, establishing a continuous link between the medical and biomechanical teams would prevent the accumulation of unsolved cases for the biomechanical team, which occurred in this study.

Complications in PRRI estimation were recorded in 7 out of 15 patients due to missing blood pressure. This was resolved using the mean cohort pressure during the study. However, patient-specific blood pressure is critical in safe/risky separation in cases where PRRI is close to the threshold (3%). This occurred to patient No. 9 (PRRI = 2.2%) and No. 15 (PRRI = 2.5%). Typically, these patients’ blood pressure should be recorded and PRRI should be updated (it would take only 2 h) but we did not conduct it due to specific circumstances here. Patient No. 15 was treated by OAR within a month and patient No. 9 refused surgery. Nevertheless, we modified our workflow for the future in order that blood pressure will be specially obtained in cases without a known blood pressure and 1% < PRRI < 7%, which is the range with the highest sensitivity of PRRI to blood pressure. Therefore, PRRI refinement has the potential to change the original decision regarding the safe/risky status of given AAA.

### 4.4. Limitations

Typically, this study has certain limitations. In addition to the small size of the study population, the fact that the study was launched after ending the state of emergency brought some limitations. Its major limitation was the biomechanical team’s overload since all cases for the assessment were received at once, which resulted in unwanted delays. Considering that one case is assessed within 3 days, this indicates that the last case was calculated 57 days after receiving the respective CT-A images. If this was a high-risk patient (PRRI > 10%), he/she would be unrecognized for almost 2 months. However, this danger can also occur in practice, especially at the beginning of the workflow. The described risk of delay can be diminished if the biomedical team classifies the received cases based on $D_{sex}$ since larger cases are more likely to be dangerous. Finally, it is acknowledged that a clinical trial is necessary in order to fully validate PRRI and its incorporation into everyday clinical practice. Until then, PRRI can be used in the decision-making process as an auxiliary toll only under non-standard circumstances, such as the current COVID-19 pandemic.

## 5. Conclusions

This study provides partial longitudinal validation of PRRI in AAA cases for the first time. Furthermore, it verifies the feasibility of incorporating BRRA in the decision-making process during the COVID-19 pandemic, in which a number of performed interventions need to be limited, in order that the healthcare system could focus more of its capacity on the treatment of patients with COVID-19 infection. However, despite the COVID-19 restrictions, we would be still be able to identify AAA cases with a high risk of rupture and offer intervention.

## Figures and Tables

**Figure 1 diagnostics-13-00132-f001:**
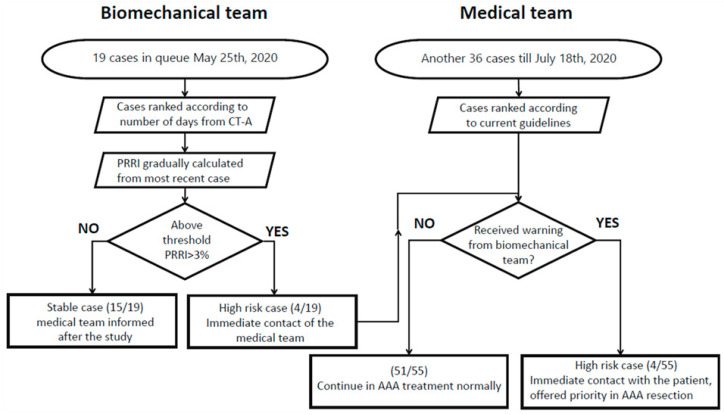
Workflow of the AAA management applied in this study.

**Figure 2 diagnostics-13-00132-f002:**
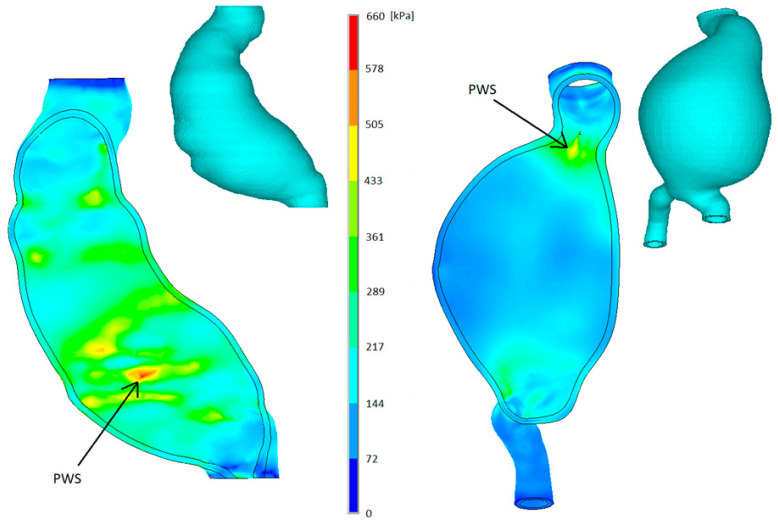
Comparison of wall stress for 2-mm-wall thick AAA geometry of the smallest high-risk case. Patient No. 16 (Female *D_sex_* = 59 mm—left) and largest safe case Patient No. 16 (Female, *D_sex_* = 87 mm—right). Arrow indicates the location of the peak wall stress (PWS). The scale is identical for both cases and AAA are suspended since the peak wall stress occurred on the inner surface. Intraluminal thrombus was not present in the left case and is not shown for the right case, although its presence was considered during computations.

**Table 1 diagnostics-13-00132-t001:** Comparison of number of cases and number of patients refusing intervention in the last 3 years. Note: In 2020, nine patients refused intervention during the COVID-19 period, one refused the intervention before this period. Rate of patients refusing interventions roughly doubled in 2020 compared to previous years.

	2018	2019	2020 (Jan–Jun)
Number of cases with $D_{sex}>55\;\mathrm{mm}$	86	82	36
Number of patients refusing intervention	9	6	10
Number of postponed interventions	-	-	10

**Table 2 diagnostics-13-00132-t002:** Clinical information of investigated patients. For patients with no blood pressure value, the available average value from the other patients was used for the BRRA analysis.

Patient No.	SEX [F/M]	AAA Diameter [mm]	Age [Years]	Blood Pressure [mmHg]	Hypertension (Treated with Medication)	Coronary Heart Disease	Diabetes Mellitus	Obesity	Hyperlipidemia	Smoking Status
1	M	60	73	164/92	yes	yes	no	yes	yes	past
2	F	49	73	N/A	no	no	no	no	yes	no
3	F	56	81	N/A	no	no	no	no	yes	current
4	M	57	76	150/75	yes	yes	no	yes	yes	never
5	M	56	72	140/70	yes	yes	no	no	yes	never
6	M	65	71	150/70	yes	no	no	no	yes	never
7	M	58	65	145/85	N/A	yes	N/A	yes	N/A	N/A
8	F	50	76	N/A	no	no	no	No	yes	N/A
9	M	56	83	N/A	N/A	N/A	N/A	N/A	N/A	N/A
10	F	82	81	130/80	yes	yes	no	N/A	yes	never
11	M	63	79	N/A	yes	ne	no	no	no	never
12	M	67	78	135/70	yes	yes	no	no	no	past
13	M	54	78	N/A	yes	no	no	yes	no	past
14	M	58	70	120/75	yes	yes	no	no	yes	yes
15	M	56	74	N/A	N/A	N/A	N/A	N/A	N/A	N/A
16	F	54	82	150/70	yes	no	no	no	no	never
17	M	57	74	120/75	yes	yes	yes	yes	yes	current
18	M	66	76	110/80	yes	yes	no	N/A	yes	N/A
19	M	83	76	130/75	yes	yes	no	No	yes	current

**Table 3 diagnostics-13-00132-t003:** Results of BRRA in the investigated cohort. * Patient died due to sepsis caused by bronchopneumonia.

Patient No.	Date of Original CTA	PRRI [%]	AAA Diameter [mm]	Follow-Up from CT-A [Days]	Reason for Follow-Up End	Outcome	False Positive Case? [Yes/No]	False Negative Case? [Yes/No]
1	19 August 2019	4.7	60	273	OAR	OAR	Yes	No
2	30 October 2019	0.8	49	217	OAR	OAR	No	No
3	26 February 2020	9.1	56	96	OAR	OAR	N/A	No
4	17 January 2020	0.8	57	136	OAR	OAR	No	No
5	13 November 2019	1	56	169	OAR	OAR	No	No
6	29 January 2020	0.1	65	570	OAR	OAR	No	No
7	25 February 2020	2.3	58	543	Study end	COVID-19 fear	No	No
8	25 July 2019	7.3	50	758	Study end	COVID-19 fear	Yes	No
9	21 March 2019	2.2	56	884	Study end	COVID-19 fear	No	No
10	10 March 2020	2.3	82	529	Study end	Refused	No	No
11	5 March 2020	0.02	63	534	Study end	Fear from surgery	No	No
12	8 January 2020	5	67	126	OAR	OAR	Yes	No
13	1 November 2017	0.3	54	980	OAR	OAR	No	No
14	28 February 2020	1.6	58	264	OAR	OAR	No	No
15	17 July 2019	2.5	56	358	OAR	OAR	No	No
16	16 April 2020	21.3	54	492	Study end	Fear from surgery	Yes	No
17	23 August 2019	0.2	57	297	OAR	OAR	No	No
18	6 March 2020	3.8	66	533	Death not AAA-related	Death due to sepsis in May 2020 *	Yes	No
19	5 January 2019	6.3	83	959	Study end	COVID-19 fear	Yes	No

## Data Availability

The data presented in this study are available on request from the corresponding author. The data are not publicly available due to the institutional GDPR policy.

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
