# Peer review of "Biomechanical Rupture Risk Assessment in Management of Patients with Abdominal Aortic Aneurysm in COVID-19 Pandemic"

_diagnostics, 2022, doi:10.3390/diagnostics13010132_

Round 1

Reviewer 1 Report

In this manuscript, authors assesses the biomechanical AAA rupture risk for patients deferred for surgical repair using patient clinical data and Finite Element Method modeling. The study was very interesting, and manuscript was well written with data supporting the conclusion. However, there were several issues which are needed to be addresses prior to its acceptance.

1. Medications have been well known to influence AAA growth rate and rupture risk, please include medications, thus it is critical to include medication, including anti-hypertension drugs, lipid-lowering drugs and anti-diabetic drug into risk assessment model.

2.  Please provide the information on whether patients have been infected with SARS-CoV-2.

3. How many time did these patients had CT aortic diameter measurements during the follow-up period?  If patient(s) had more than two diameter measurements, please provide data on aneurysm enlargement rate (monthly or annual). 

Author Response

We would like to thank the reviewer for their time and effort to closely read and review our manuscript. We appreciate all the comments and we did our best to answer reviewer´s questions and update the manuscript accordingly to improve its comprehensibility.

  1. Medications have been well known to influence AAA growth rate and rupture risk, please include medications, thus it is critical to include medication, including anti-hypertension drugs, lipid-lowering drugs and anti-diabetic drug into risk assessment model.
  • We thank the reviewers for this comment. According to current clinical guidelines effect of medication on AAA growth has not been proven so far (Wanhainen et al. 2019). Mentioned medication is intended to reduce cardiovascular risk not risk of AAA rupture. And the computational model we used calculate only risk of AAA rupture at given geometry (Polzer and Gasser 2015, Polzer et al. 2020) so it is valid as long as the geometry of AAA does not change significantly. Further, inclusion of any new parameter into rupture risk assessment model would need to be validated first retrospectively as we currently had not data supporting inclusion of medication into rupture risk model. Nevertheless, we modified Table 2 to specify if patients used antihypertensive medication to satisfy Reviewer´s request.  

  1. Please provide the information on whether patients have been infected with SARS-CoV-2.
  • According to our hospital database none of the patients included in this project have been infected with SARS-CoC-2 during the follow up. We include this information into the manuscript. Line 253-255

  1. How many time did these patients had CT aortic diameter measurements during the follow-up period?  If patient(s) had more than two diameter measurements, please provide data on aneurysm enlargement rate (monthly or annual). 
  • Two patients (2) had second CT scan during follow up – growth rate for the first one was 10 mm / year, for second one it was 7.6 mm / year. All other patients had only one baseline diameter measurement during the follow up period. The information about the AAA growth is now included into manuscript. Line 236-238

Reviewer 2 Report

Dear Author, 

Your Manuscript validating a new method to assess the probability of aneurysm rupture, using the Covid-19 pandemic years as the temporal window, is original and interesting. 

However some limitations are present, as you assessed in the final paragraph, and I think you should clarify some points of your research.

Line 48: "the majority of patients are not under imminent risk of rupture when operated on" please note some bibliographical notes. otherwise it is just your opinion. 

Line 155: "The PRRI was computed by the biomechanical team as described in chapter 2.2": why do you decide to use this method? How has this index been developed? How has exactly done? please specify better. 

Line 144-146: "in this situation the average value from other patients was used during the analysis. Absence of valid blood pressure may be significant limitation in PRRI estimation if its value obtained by mean cohort pressure is close to the safe/risky threshold" The absence of the blood pressure is a significant limitation. However I was not able to find how many  patients' pressure was missing. And does it affect the result. (not only  in the case described at the end as high risk)

Line 187: "This approach does not assess peak wall stress from Laplace law. Instead, the Finite Element (FE) method divides any geometry into a large number of small FEs" Why does this method not use the Laplace law? Why did you decide to use this method? Can you please give us an overview of your method choice? 

Line 197: "respects zero pressure geometry of AAA": is the zero pressure geometry the only geometry usable? Are there any alternatives? Please in the discussion session, explain better the characteristics of your model. 

Line 199: "peak wall stress evaluated when AAA overloaded by 1.5x patient-specific mean arterial pressure" is this pressure threshold useful? Why has been choosen? Why not x1.4 or 1.6 pressure? Is it possible to stratify the risk with other pressure threshold? 

Line 203: "which  is, for the modelling purpose, assumed to occur once a year." Is it possible to stratify for more than once a year? Please explain why you choose this method, if you think there are some limitation or if you think it is a good method as it is. 

Line 211-212:"PRRI was able to decrease the ratio of false positive cases to one third compared to ???? but its power decreases with increasing time from the initial CT-A." How do you assess it? which is the statistical analysis used? Is it statistically significant? Which test did you use? 

Line 214: "CT-A no older than 9 months" Why did you choose 9 months? Please explain

Line 209-228: this paragraph should be moved to the discussione section

Line 236-246: "while 6 cases were identified as high risk (PRRI>3%)": two patients were operated and two refused. What about the other two? 

Line 244: "other eight still refuse the repair and are followed, all of them are alive at the time of the paper creation (August 2021)" How long was the follow-up lenght of this patients? 

Line 301: "the BRRA is superior, providing a decrease of necessary intervention by at least 67% and likely to reach 80%" How can you say that? How did you calculate that? Please explain better. 

Line 325: "the overwhelming majority of cases (92%) with this PRRI should remain intact for one year." No rupture cases are reported in your study, how can you express false positive in this contest? 

Line 345: "his study suggests to postpone interventions in large AAA cases (up to 70 mm) of older patients in the environment of significant probability of COVID-19 infection." How can you say that? Don't you think that your small sample to assess such a great statement? Have you done a statistical test to assess it? 

Author Response

We would like to thank the reviewer for their time and effort to closely read and review our manuscript. We appreciate all the comments and we did our best to answer reviewer´s questions and update the manuscript accordingly to improve its comprehensibility.

Line 48: "the majority of patients are not under imminent risk of rupture when operated on" please note some bibliographical notes. otherwise it is just your opinion. 

  • citations added.

Line 155: "The PRRI was computed by the biomechanical team as described in chapter 2.2": why do you decide to use this method? How has this index been developed? How has exactly done? please specify better. 

  • We thank the reviewer for this comment. There is a typo in mentioned sentence as PRRI is described in chapter 2.3 not 2.2. We believe the chapter 2.3. explains PRRI briefly but sufficiently and readers are referred to study Polzer and Gasser 2015 where PRRI was developed and it is described in details. This index was chosen because there was urgent need to decrease number of operations during covid pandemic and we previously shown (Polzer et al. 2020) PRRI has potential do decrease number of operations by some 50% without compromising sensitivity of currently used Dmax (i.e. without missing any case which would rupture).  

Line 144-146: "in this situation the average value from other patients was used during the analysis. Absence of valid blood pressure may be significant limitation in PRRI estimation if its value obtained by mean cohort pressure is close to the safe/risky threshold" The absence of the blood pressure is a significant limitation. However I was not able to find how many  patients' pressure was missing. And does it affect the result. (not only  in the case described at the end as high risk)

  • This information is specified in Table 2. It was not known for 7 patients and the consequences are explained in the last paragraph of chapter 4.3. It could not alternate outcome of this study due to specific circumstances (patient 15 was operated in month and patient 9 refused intervention anyway regardless of his PRRI. Nevertheless we are aware of this limitation and we modified our workflow to prevent this in the future as specified in the las paragraph of section 4.3.

Line 187: "This approach does not assess peak wall stress from Laplace law. Instead, the Finite Element (FE) method divides any geometry into a large number of small FEs" Why does this method not use the Laplace law? Why did you decide to use this method? Can you please give us an overview of your method choice? 

  • Laplace law relies on several assumptions which are not fulfilled in case of AAA. 1st. it can estimate only mean stress across wall thickness thus it cannot correctly describe response under elevated BP and secondly it is not correct in cases the wall curvature is comparable with its thickness (i.e. in areas of local protrusion or bulges where highest stress often occurs) and under thick ILT. This problem has been extensively discussed in our previous papers Polzer et al. 2013 and Man et al. 2018 and in Polzer and Gasser 2015. Finally, it is noted this study did not aim to assess strength of several criterions against each other. Instead we aimed at prospective validation of PRRI which had the best results previously (see Polzer et al. 2020).

Line 197: "respects zero pressure geometry of AAA": is the zero pressure geometry the only geometry usable? Are there any alternatives? Please in the discussion session, explain better the characteristics of your model. 

  • The zero pressure geometry should be used when realistic material model is used. Without it the deformed geometry would not be the one we see on CT, thus it would not be realistic. The discussion was modified accordingly. Line 198-200

Line 199: "peak wall stress evaluated when AAA overloaded by 1.5x patient-specific mean arterial pressure" is this pressure threshold useful? Why has been choosen? Why not x1.4 or 1.6 pressure? Is it possible to stratify the risk with other pressure threshold? 

  • We admit this is somehow arbitrary value which was intended to mimic rare but still realistically high value. That choice however was done in Polzer et al. 2015 and we kept it through our retrospective validation so we are not going to change during the prospective validation as that would reset the whole process. More importantly, we do not have any value which would be more reasonable. And the manuscript clearly states that more advanced rupture risk estimation (Polzer et al. 2021) requires PS holter measurements which were not available. That is why we applied our older yet more validated PRRI criterion.
  • Regarding the second part of your question. We showed the Wall stress depend almost linearly on BP (see Fig.3 in Polzer et al. 2021). Therefore using of other sufficiently high BP value would change the PRRI but not the ranking of patients. Most importantly in that case we would not have a criterion for separation risky from safe patients. That was estimated in Polzer et al. 2020 to 3% using 1.5xMAP and that is why we need to keep this value also in this study.

Line 203: "which is, for the modelling purpose, assumed to occur once a year." Is it possible to stratify for more than once a year? Please explain why you choose this method, if you think there are some limitation or if you think it is a good method as it is. 

  • The reviewer is right this was not correctly written. The result is the same even if the 1x5MAP would occur every heartbeat. Correct sentence now says “which, for modelling purposes, is assumed not to be exceeded in a year.
  • All the assumptions described here were made back in definition of the PRRI criterion (Polzer et al. 2015) and this study aims only on its validation based on previous results. That is why we did not modify it anyhow. Any way to answer Reviewers question it I based on idea the AAA will rupture under the highest pressure which can occur in a year. That gives the PRRI meaning comparable to annual risk of rupture known to clinicians.

Line 211-212:"PRRI was able to decrease the ratio of false positive cases to one third compared to ???? but its power decreases with increasing time from the initial CT-A." How do you assess it? which is the statistical analysis used? Is it statistically significant? Which test did you use? 

  • This is only brief description of results of our previous study Polzer et al. 2020 (reference 10). Yes, there were one tailed Man Whitney statistical tests used to reveal that (see Table I in ref. 10) 

Line 214: "CT-A no older than 9 months" Why did you choose 9 months? Please explain

  • We believe we explained that clearly in the very next sentence:

“That is why we applied it first to cases with CT-A no older than 9 months, which is the limit until which PRRI gives a more accurate prediction than  [10].

Anyway we reworded it to be even more clear.

Line 209-228: this paragraph should be moved to the discussione section

  • The Reviewer is right. We moved part of the paragraph in to discussion as suggested. We only kept the part where results from We previously published study [10] is described and also the part where we explain how we rank cases to solve.

Line 236-246: "while 6 cases were identified as high risk (PRRI>3%)": two patients were operated and two refused. What about the other two? 

  • There is a typo this should be “three patients were operated and remaining four refused”

Corrected in manuscript

Line 244: "other eight still refuse the repair and are followed, all of them are alive at the time of the paper creation (August 2021)" How long was the follow-up lenght of this patients? 

  • This information can be found in Table 3 – the length of follow up was 543, 758, 884, 529, 534, 492, 533 and 959 days for each respective patient.

Line 301: "the BRRA is superior, providing a decrease of necessary intervention by at least 67% and likely to reach 80%" How can you say that? How did you calculate that? Please explain better. 

  • We are thankful for this comment as those numbers were not correct. We in fact, re think our definition of false positive cases and make it more benevolent (i.e. the outcome is now slightly worse for our PRRI index). The manuscript was changed accordingly in this section as well as in Table 3.

Line 325: "the overwhelming majority of cases (92%) with this PRRI should remain intact for one year." No rupture cases are reported in your study, how can you express false positive in this contest? 

  • False positive in our context means cases with high PRRI with higher probability of imminent rupture (according to our calculation) “positive”, but no rupture occurred during follow up “false”. This method is still calculating with probability, it means that even if we found an AAA to be in high risk of rupture doesn´t mean that the rupture is certain in set period of time, it just says that the probability of AAA rupture is higher than we would estimate from maximal diameter criterion.

Line 345: "his study suggests to postpone interventions in large AAA cases (up to 70 mm) of older patients in the environment of significant probability of COVID-19 infection." How can you say that? Don't you think that your small sample to assess such a great statement? Have you done a statistical test to assess it? 

  • We are thankful for this comment as this was our bad wording. The postponing of all AAAs during covid waves is not our suggestion but it is recommendation of American College of Surgeons [ref. 23]. We only point out this will lead to increase of ruptures while our studies (both this one and previous study [10] suggests postponing of cases based on BRRA would allow decrease of interventions by two thirds without increase of ruptures. The manuscript has been rewritten accordingly.